# The Influence of Increased Pedicle Screw Diameter and Thicker Rods on Surgical Results in Adolescents Undergoing Posterior Spinal Fusion for Idiopathic Scoliosis

**DOI:** 10.3390/jcm13082174

**Published:** 2024-04-10

**Authors:** Pawel Grabala, Piotr Kowalski, Michal Grabala

**Affiliations:** 1Department of Pediatric Orthopedic Surgery and Traumatology, Medical University of Bialystok and Medical University of Bialystok Children’s Clinical Hospital, ul. Waszyngtona 17, 15-274 Bialystok, Poland; 2Paley European Institute, Al. Rzeczypospolitej 1, 02-972 Warsaw, Poland; 3Department of Neurosurgery with Department of Interventional Neurology, Medical University of Bialystok and Medical University of Bialystok Clinical Hospital, ul. M. Sklodowskiej-Curie 24A, 15-276 Bialystok, Poland; 4Department of Neurosurgery, Regional Specialized Hospital, ul. Dekerta 1, 66-400 Gorzow, Poland; pkowal72@gmail.com; 52nd Clinical Department of General and Gastroenterogical Surgery, Medical University of Bialystok and Medical University of Bialystok Clinical Hospital, ul. M. Skłodowskiej-Curie 24A, 15-276 Bialystok, Poland; michal@grabala.pl

**Keywords:** adolescent idiopathic scoliosis, AIS, idiopathic scoliosis, scoliosis, spinal deformity, posterior spinal fusion, PSF, pedicle screws, thicker rods

## Abstract

**Background:** Modern surgical techniques allow for the correction of spinal deformity, stopping its progression and improving pain relief and social and physical functioning. These instruments have different implant designs, screws, and rod diameters and can be composed of different metal alloys with different hardnesses, which can have a significant impact on the effect of correcting spinal deformities. We designed a retrospective cohort study based on the same surgical technique and spine system using different implant sizes, and compared the results across them. **Methods:** This is a retrospective review of adolescent idiopathic scoliosis (AIS) patients who underwent posterior spinal fusion (PSF) between 2016 and 2022 with a minimum two-year follow-up (FU) using two spinal implant systems: 5.5 and 6.0 mm diameter screws with double 5.5 mm titanium rods (Group 1 (G1)), and 6.0 and 6.5 mm diameter pedicle screws with double 6.0 mm cobalt–chromium rods (Group 2 (G2)). The evaluated data were as follows: preoperative personal data, radiographic outcomes, complications, and health-related quality of life questionnaire (HRQoL). The parameters were reviewed preoperatively, after the final fusion, and during the FU. **Results:** The mean age of all 260 patients at surgery was 14.8 years. The average BMI was also similar in both groups and was noted as 21. The mean levels of fusion and screw density were similar in both groups. The mean preoperative major curves (MCs) were 57.6° and 62.5° in G1 and G2, respectively. The mean flexibility of the curves was noted as 35% in G1 and 33% in G2. After definitive surgery, the mean percentage correction of the MC was better in G2 vs. G1, with 74.5% vs. 69.8%, respectively (*p* < 0.001). At the final FU, the average loss of correction was 5.9° for G1 and 3.2° for G2 (*p* < 0.001). The mean preoperative (TK) thoracic kyphosis (T2–T5) was 12.2° in G1 and 10.8° in G2. It was corrected to 15.2° in G1 and to 13° in G2. At the FFU, we noted a significant difference in the TK (T2–T5) between the groups, with 16.7° vs. 9.6° for G1 vs. G2, respectively (*p* < 0.001). Statistical significance was observed between the preoperative sagittal balance and the final follow-up for both groups (*p* < 0.001). **Conclusions:** AIS patients surgically treated with screws with a larger diameter and thicker and stiffer rods showed greater correction and postoperative thoracic kyphosis without implant failure. The complication rates, implant density, and clinical outcomes remained similar. The radiographic benefits reported in this cohort study suggest that large-sized screws and stiffer rods for the correction of pediatric spinal deformities are safe and very effective.

## 1. Introduction

Most spinal deformities in children and adolescents have unknown etiology and are, therefore, called idiopathic. Adolescent idiopathic scoliosis is defined as a three-plane deformation of the spine with a Cobb angle of more than 10 degrees, measured on a standing radiograph of the entire spine. It affects approximately 1–3% of adolescents, and the incidence is similar in men and women; however, women are 10 times more likely to develop a Cobb angle of 30 degrees or more [1,2,3]. Curvatures that reach 50 degrees despite conservative treatment require surgical treatment [3,4]. Severe curves can cause disabling pain and fatigue, decreased respiratory capacity, cardiopulmonary symptoms, and high rates of mortality and morbidity [5,6,7]. With the ubiquity of medicine, spine surgery has also been developed. Modern surgical techniques allow for the correction of the curvature, stopping its progression and improving pain relief and social and physical functioning [8,9,10,11,12,13]. Since the introduction of Cotrel–Dubousset instrumentation, it has become the “gold standard” in the treatment of idiopathic scoliosis, with many modifications of implants and instruments allowing for the optimal correction of deformities [10,14]. However, these instruments have different implant designs, screws, and rod sizes and can be composed of different metal alloys with different hardnesses and rod thickness, which can have a significant impact on the effect of correcting spinal deformities [15,16,17,18,19,20,21,22,23].

There are few studies on the results of the surgical treatment of AIS based on the same instrumentation system using different sizes of rods, screws, and rod types [15,16,17,18,19,20,21]. To find out whether the rod size, rod type (metal alloy), and screw size influence surgical outcomes in AIS, we designed a retrospective cohort study based on the same spine system with different implant sizes and compared the results across them.

## 2. Materials and Methods

### 2.1. Setting and Patients

This study is an IRB-approved retrospective review of AIS patients who underwent PSF between 2016 and 2022 at medical spine centers due to the affiliations of the authors (P.G., P.K.) with a minimum two-year follow-up (APK.002.78.2020). All of the patients and their parents provided written consent relating to their participation in the study and publication of the results. From January 2016 to December 2022, 260 consecutive patients with AIS were treated surgically with segmental screw instrumentation via a posterior approach at our hospital. The indications of surgery were a Cobb angle of more than 50 degrees with progression in skeletally immature patients, and a Cobb angle of more than 50 degrees or with back pain in skeletally mature patients younger than 18 years, but not more than 90 degrees. The inclusion criterion for the study was surgical treatment of AIS. All patients underwent surgical treatment in one stage, without any additional procedures, like halo gravity traction or three-column osteotomies. Patients who had previous back surgery or anterior surgery, who underwent revisions, or who had spinal deformities of different etiology were excluded from the analysis. All surgeries were performed by the same senior spine surgeons using the same freehand technique for screw placement and surgical dissection and instrumentation procedures. All studied cases were performed using two spinal implant systems: 5.5 and 6.0 mm diameter screws with 5.5 mm titanium rods (Group 1 (G1)), and 6.0 and 6.5 mm diameter pedicle screws with 6.0 mm cobalt–chromium rods (Group 2 (G2)). Both larger screws and standard screws were utilized at all levels, thoracic and lumbar, with similar screw density. The preoperative personal data, including sex, age, Risser grades, and body mass index (BMI), were recorded from medical records.

### 2.2. Outcome Parameters

In the studied patient groups, we assessed the following parameters: radiological examination results, perioperative complications, and health-related quality of life (HRQoL) using the SRS-22r questionnaire. We recorded all assessed parameters preoperatively, after definitive surgery, and during the follow-up [7,24,25]. Complications in both groups of patients, if they occurred, were also analyzed.

### 2.3. Radiological Measures

Standard standing posteroanterior and lateral radiographs of the whole spine and side-bending films were obtained for all patients before surgery. At each follow-up visit after surgery, postoperative standing posteroanterior and lateral radiographs of the whole spine were performed. Objective parameters were documented and used to compare the differences between the G1 and G2 instrumentation, including coronal measure of curvature, thoracic kyphosis, flexibility of the main curvature, and coronal global balance. We also analyzed operative time, and estimated blood loss. The intraoperative and perioperative outcomes were reviewed. The curves were classified according to the Lenke classification [26]. The Cobb method was used for measuring coronal curves with a standing triple film of the whole spine. The flexibility of the main curvature was analyzed on a side-bending film. The coronal global balance (CGB) of the curve was marked and noted using the distance between the C7 plumb line and the center sacral vertical line. The Cobb angles of all curves were noted, and sagittal measurements—thoracic kyphosis (T5–T12) and lumbar lordosis (T12–S1)—were also included. The correction percentages of the main curves were then calculated. All measurements from radiographs were taken by an independent observer. All patients treated surgically underwent an MRI examination of the entire spine performed before surgery to exclude other pathology of the spinal cord. During surgery, all patients underwent intraoperative spinal cord monitoring, including somatosensory evoked potentials (SSEPs) and transcranial motor evoked potentials (MEPs) [27,28].

### 2.4. Surgical Technique

All the operations, which consisted of posterior correction with CD instrumentation and fusion with autogenous bone graft, were carried out by the same two senior orthopedic surgeons (P.G. and P.K.). The same surgical technique was used for all patients. We performed corrections of spinal deformities via the posterior approach using segmental screw instrumentation [10]. The patients were placed in the prone position after induction of general anesthesia with tracheal intubation. The standard posterior approach was used for access to all planed fixation levels. Pedicle screw placement was performed via free-hand technique due to steps described in the literature by Suk and Lenke [29,30,31,32]. Facetectomies, pedicle screw placement, with posterior column osteotomies (Ponte), was typically performed at several levels on the apex of the curvature [33,34,35,36]. Next, two rods were measured, cut, and contoured for coronal and sagittal plane alignment. The rods were inserted, connected to the pedicle screws, and double-rod synchronic derotation was performed with neuromonitoring [27,28]. Next, in a safe and optimal manner, the deformity was corrected using a combination of rod cantilevering and derotation. Decortication was performed at all levels of the planned fusion, and allograft bone was placed in a posterolateral fashion. The wound was then closed in layers over subfascial drains. The patients were allowed to sit in bed at 24 h postoperatively. The drains were removed at 48 h postoperatively, and patients were allowed to exercise and walk. No postoperative immobilization was used. The patients were allowed and encouraged to ambulate at postoperative day 5, on average, or as soon as they could tolerate it. Every patient was followed up at our clinic at postoperative intervals of 3 weeks, 3 months, 6 months, 12 months, and then annually.

### 2.5. Statistical Considerations

In our study, we used Statistica statistical analysis software (version 10.0; StatSoft Inc., Tulsa, OK, USA) for all analyses. ANOVA and the Tukey–Kramer method were used. For the analysis and presentation of the data, we used the standard deviation (SD) as the mean, and independent *t*-tests and a Mann–Whitney test was used to analyze the continuous data, while the categorical variables, including the complication rates, were analyzed utilizing Fisher’s exact and the chi-square tests, and displayed as the frequency and percentages. The chi-square test and Fisher’s exact test were used for parametric and non-parametric data when appropriate. For variables having negative or positive values based on the measured reference point, such as coronal balance, statistical comparisons of groups required converting negative numbers to positive numbers because of the necessity to statistically analyze differences from a reference point. The *p*-value considered statistically significant was set at less than 0.05 before analysis.

## 3. Results

The mean age of all 260 patients at surgery was 14.8 years in both groups. The mean BMI was also similar in both groups and noted as 21. The mean Risser grade was 3.6 for G1 and 3.2 for G2. No statistical significance was observed in the demographic data (as shown in Table 1), including age, sex, BMI, and Risser grade, in the classification of the curve patterns between the groups (N.S.).

### 3.1. Clinical Characteristics and Radiographic Outcomes

The mean (SD) operative times were recorded as 252 (62.4) minutes for the G1 patients and 344 (82.8) min for the patients in G2. The mean (SD) estimated blood loss levels were noted as 480 (288) mL for the patients in G1 and 632 (278) mL for G2. During statistical analysis, we found no statistically significant differences in either operative time or estimated blood loss between the study groups. The mean (SD) fusion levels were 10.2 (2.8) for G1, and 9.8 (3.2) for G2 (N.S.). The mean (SD) screw densities were noted as 75% (12) for the G1 group, and 75 (10) for the G2 group (N.S.). All analyzed data are presented in Table 1. Figure 1, Figure 2, Figure 3 and Figure 4 show X-rays of exemplary patients with idiopathic scoliosis after surgical treatment from G1 and G2.

The mean (SD) preoperative major curves (MCs) were 57.6° (11.9) and 62.5° (12.1) in G1 and G2, respectively (N.S). The mean preoperative (SD) flexibility of the curvatures analyzed in the bending films was recorded as 35% (11) in G1 and 33% (8) in G2. After definitive surgery, the MCs were corrected to 19.8° (10.2) and 16.6° (8.2) in G1 and G2, respectively (*p* < 0.001). Comparing preoperative and postoperative Cobb angle measurements of the frontal curve, the mean (SD) percentage of MC correction was better in G2 compared to G1, 74.5% (11.2) vs. 69.8% (12.8), respectively (*p* < 0.001). At the final follow-up, the average (SD) loss of correction was 5.9° (3.6) for G1 and 3.2° (3.2) for G2 (*p* < 0.001). Figure 5 shows the main curve measures in both groups.

However, in the analysis we observed a statistical difference in the final correction rate in the frontal plane between patients in G1 and G2. At the FFU, the MC (SD) was measured at 21.3° (9.2) for the SG group and at 17.1° (7.2) for the LG group (*p* < 0.001). All analyzed radiographic measures are presented in Table 2.

The mean (SD) preoperative thoracic kyphosis (T2–T5) degrees were 12.2° (8.8) in G1 and 10.8° (8.9) in G2 (N.S.). They were corrected to 15.2° (9.8) in G1 and 13° (5.9) in G2 (N.S.). The mean (SD) preoperative thoracic kyphosis (TK) (T5–T12) degrees were 25.1° (15.5) in G1 and 26.9° (15.8) in G2 (N.S.). They were corrected to 24.8° (8.8) in G1 and 23.5° (10.8) in G2 (N.S.). The mean (SD) preoperative thoracic kyphosis (T2-T12) degrees were 31.2° (14.8) in G1 and 33.5° (13.6) in G2 (N.S.). They were corrected to 33.1° (11.8) in G1 and 31.4° (12.6) in G2 (N.S.). At the FFU, we noted a significant difference only in the thoracic kyphosis (T2–T5) degree between the groups, with 16.7° (9.2) vs. 9.6° (6.8) for G1 vs. G2, respectively (*p* < 0.001).

Figure 6 and Figure 7 show the thoracic kyphosis values in both groups during the treatment course.

The mean preoperative lumbar lordosis degrees were 50.8° (10.2) in G1 and 50.2° (9.6) in G2, which were corrected to 44.2° (12.4) in G1 and 46.4° (11.8) in G2 (N.S.) and noted at FFU to be 50.2° (11.8) in G1 and 49.5° (12.3) in G2 (N.S.).

In terms of the CGB measurement, the mean (SD) value was 18.5 (11.4) mm for G1 and 19.2 (11.8) mm for G2. After surgery, the mean (SD) distances were 8.8 (5.6) mm for G1 and 8.1 (6.9) mm for G2. At the final follow-up, the mean (SD) distances were 9.8 (6.2) mm for G1 and 10.6 (7.4) mm for G2. In the statistical analysis, we did not note any statistical significance in the GCB between these two groups. Statistical significance was recorded between sagittal balance before surgery during the final follow-up in both groups (*p* < 0.001).

### 3.2. HRQoL

During the follow-up period, patients experienced a marked and significant improvement in the mean total score of the preoperative SRS-22r assessed questionnaire from 3.86 to 4.36 in G1 and from 3.82 to 4.26 in G2 (*p* < 0.001 for both comparisons) (Table 3). Thanks to a thorough analysis of the SRS-22r questionnaire, we noted a clinically and statistically significant improvement in self-image and satisfaction parameters in both groups compared to preoperative and final follow-up results (*p* < 0.001). At the last follow-up visit, the mean total score improved significantly in both groups.

### 3.3. Complications

The patients we analyzed in this study experienced intraoperative and postoperative complications, as shown in Table 4. In both groups analyzed, we noted that postoperative complications occurred in 19.7% of patients in group G1 and 23.5% of patients in group G2. None of the patients obtained a new postoperative neurological deficit in G1 or G2. A total of 3.8% of patients in G1 and 4.7% in G2 underwent revision surgery (screw replacement or reconstruction of dura). No complications were reported at the final follow-up.

## 4. Discussion

The correction of spinal deformities using modern systems based on the CD technique enriched with osteotomy techniques of the posterior column of the spine provides powerful opportunities for the correction and stabilization of curvatures [4,8,10,11,12,13]. Many details of implants have evolved and been improved, such as the implant profile, screw head stiffness, mobility in the sagittal axis, and its multi-axial nature, to obtain the best possible correction results [20,21,37,38]. The quality of the rod used is important in three-plane correction, i.e., the metal alloy (titanium, cobalt–chrome) and its diameter, which affects its ability to reproduce the sagittal and frontal balance of the spine [17,18,19,20,21]. It is known from other biomechanical studies that the larger the screw diameter, the better the stabilization and the lower the risk of screw removal and implant loosening [20,23,39,40,41]. For the surgical treatment of AIS, we received excellent outcomes using a system based on CD instrumentation in terms of three-dimensional correction of the spine (Figure 1, Figure 2, Figure 3 and Figure 4 show X-rays of exemplary patients with idiopathic scoliosis after surgical treatment from G1 and G2), the coronal and sagittal planes, derotation, and the clinical outcomes [11,12,13]. It should be emphasized that the correction possibilities of spinal deformation are influenced by other factors that we did not analyze, such as bone quality, so we focused on assessing the results of surgical treatment based on the sizes of the screws and rods used [39,40,41]. Also, the flexibility of the spine assessed on bending films is the main factor influencing the achievement of correction, as well as the loss of correction [42].

Based on our retrospective study, it can be clearly stated that in patients surgically treated for spinal deformity who were treated with larger screw sizes connected to thicker and harder rods, greater overall correction of the coronal curve, better restoration of thoracic kyphosis, and better derotation of the apex were observed. In our series, we noted that the percentage of coronal curve correction was greater in the G2 patients with larger screws and thicker rods. In our study, the patients in G2 who underwent PSF received significantly better coronal Cobb correction (74.5% vs. 69.8%, *p* < 0.001) than the standard screw patients, and the apical vertebral translation at the final follow-up was significantly better in G2. Our reports are comparable to previous reports [20,21]. The main goal of using implants in spine surgery is to obtain three-plane deformation correction and then maintain the correction until fusion is achieved as well as postoperative mobilization as soon as possible. Segmental screw constructs allow for stabilization of the spine in all three columns, improving fusion by immobilizing instrumented segments and providing greater corrective forces [20,21,43,44,45]. The stiffness of the spinal implants is influenced by the geometric properties of the instrumentation, including their size and shape, as well as the properties and structure of the material from which the implants are made [38,43]. Biomechanically, the size of the rod in proportion to its diameter and the material properties may influence the stiffness of the implants [38,46,47,48]. This finding is not surprising, given that pedicle screws with an increased diameter and the concomitant increased biomechanical strength resulted in greater correction. The strength of the structure used in spinal correction depends primarily on the material and mechanical properties of the pedicle screw’s material and mechanical properties, with a larger diameter significantly increasing strength [23,46,47,48,49,50,51,52,53]. In the study by [47], the authors concluded that the screw size significantly affected fixation stiffness, with larger screws increasing the stiffness [54]. This possibly explains the outcomes of spinal deformity correction, with improvements in thoracic kyphosis and coronal correction in this study. We noted at the final follow-up that the mean percentages of coronal curve correction were 61% in G1 and 68% in G2, with mean losses of correction of 5.9 degrees in G1 and 3.2 degrees in G2. We observed a statistical difference between these two groups, including the postoperative Cobb angle at the FFU, the percentage of correction achieved, and the final percentage of correction and loss of correction at the FFU (*p* < 0.001). No statistical differences were noted in the flexibility or fusion levels, Ponte osteotomy, or lumbar lordosis measures. A primary objective for spinal fusion in scoliosis patients is to restore the sagittal balance [47,48,54,55]. Pedicles are the strongest part of the vertebra, providing the strongest fixation point for the spine; however, there are mixed results in achieving normal postoperative kyphosis after pedicle screw instrumentation [46,56]. Based on the available scientific research, it has been proven that performing Ponte osteotomies (POs) is a direction to counteract this sagittal flattening. By removing the posterior ligaments and joints that act as bonds, we can increase the flexibility of the spine and obtain better possibilities of spine correction in three planes [23,33,55]. Mostly hypokyphotic AIS curves achieve posterior shortening of the thoracic spine, allowing for the restoration of kyphosis [30,33,57]. Our study showed significantly better restoration in thoracic kyphosis at the FFU. We noted a significant difference in thoracic kyphosis (T2-T5) between the groups, with 16.7° vs. 9.6° for G1 vs. G2, respectively (*p* < 0.001). Based on the available research, a thicker rod has been shown to be stiffer, and provides better correction and curvature control; however, a thinner rod has better flexibility and plasticity, which ensures ease of use during implantation and curvature correction. However, on the other hand, a thinner and softer rod cannot cope with the correction of more rigid curvatures and deformities during spine correction maneuvers, which results in weaker correction and sometimes the loss of thoracic kyphosis and the effect of a straight back. The available medical literature has often confirmed the greater stiffness of the structure, which was at least partially dependent on the diameter of the structure rod [58]. By using a thicker and stiffer rod, we are able to obtain greater corrective forces on the deformed spine, and in combination with larger sizes of pedicle screws and Ponte osteotomy of the posterior column of the spine, we will obtain the most desirable correction in three planes, as our study shows [22,23,30,57]. This trend persisted even after separating patients by curve flexibility. As the spine translates coronally and sagitally to a contoured rod, it also derotates because of the force vector. Therefore, a stronger fixation secondary to larger diameter screws increases the corrective forces applied to the deformed spine in all three planes, allowing for increased correction in all three dimensions. However, recent studies have shown that POs may have little to no impact on the sagittal plane and may be associated with neuromonitoring complications, higher costs, and risk of readmission, without a benefit in patient-reported outcomes [34,35,36].

This higher kyphosis can be explained by the increased fixation strength of larger-diameter screws, which can maintain the contoured rod bend and thoracic curvature during rotation [34,36,57]. Matsukawa et al. conducted a finite element analysis comparing the influence of the segmental screw diameter, length, and fill on construction in osteoporotic vertebrae and found that an increased screw diameter was significantly more resistant to vertebral flexion–extension loading [50]. As the fixation strength of the pedicle screws surpasses spinal loading due to the increased screw diameter, the surgeon has more control in maintaining normal sagittal alignment. Additionally, overall complication rates in our study were similar in both groups. Only intraoperative neuromonitoring changes were higher in G2 vs. G1, with 3.8% vs. 5.5%, respectively (<0.001).

Aside from correction, safety and accuracy remain top priorities in scoliotic instrumentation. Segmental screw placement in the upper- and mid-thoracic spine is technically challenging, with high misplacement rates [51,59,60]. In particular, increased screw diameters are often avoided for pediatric thoracic vertebrae, given their smaller pediatric pedicle size [22,59,61]. In the study by Cho et al., the authors concluded that while large screw diameters (up to 9.5 mm) caused an increase in the pedicle circumference, there was no spinal canal compression [52]. Additionally, 99.3% of screw breaches in their study were lateral [52]. In the study by Sarwahi et al. [21], only one (2.0%) large-screw patient had a lateral breach on their CT and no incidence of medial breaching. This rate is lower than the 5.0–15.7% rates of misplacement previously reported [53,61,62,63,64]. All three misplacements in their study were lateral, within the acceptable range of breach, or the costovertebral complex [52,65,66,67]. In one of the largest studies assessing the accuracy of pedicle screw placement in the pediatric population via the freehand technique, the authors examined 6358 screws implanted in vertebrae during scoliosis correction and found that 98% of the screws were placed correctly. Also in this study, only 0.88% breached more than 4 mm vertebral walls and 0.26% of the screws were re-inserted. There were no new neurological, vascular, or visceral complications [60]. This was possible because the medial wall is 2–3 times thicker than the lateral wall at the thoracic level, which resists medial breaching [22,56]. However, in another study of screw placement, analyzing the increased diameter of the vertebral pedicle screw in the pediatric population, very interesting results were obtained that there is a wide range of vertebral pedicle expansion during screw insertion (up to 78%), with a low risk of lateral or medial fracture and no increased risk of complications [22], which is confirmed and justified by other previous biomechanical studies [52,66,67]. The larger the diameter of the screw inserted into the pedicle, the more the pedicle expands [22,52,66,67]. Additionally, previous studies analyzing pedicle screw size have been on cadavers or adult spine deformities. In comparison, pediatric pedicles are more elastic, increasing the ability of these pedicles to accommodate larger-diameter screws [52,66,67]. Continuous improvements in surgical navigation can also minimize the risk of severe misplacements. In a study by [57], it was observed that in patients with a hypokyphotic thoracic spine, a significant positive correlation was obtained between TK change and multilevel facetectomy or screw density on the concave side. However, in the large review the authors [23] showed better correction capabilities when using more stiff cobalt–chrome rods. Taking these observations into account, we believe that our study accurately confirms the better correction options for spinal deformities, especially the reconstruction of thoracic kyphosis and coronal correction by using larger sizes of screws [22] and rods combined with multi-level spinal mobilization [33,34,35] as shown by our studies. We know that the correction of spinal deformity is influenced by many factors, both surgical and biomechanical. Identifying these factors and improving surgical techniques will yield the best surgical results.

In the studied groups of patients, no neurological complications occurred, and intraoperative NM changes were not related to the penetration of the screw into the spinal canal or collision with the spinal cord. Also, cerebrospinal fluid leakage did not occur after screw insertion, but was revealed during the preparation of the screw canal. NM changes were not complete and did not exceed a decrease of more than 50% of the baseline; they occurred during the derotation maneuver and after raising blood pressure above 100 mmHg, after which all NM potentials returned to normal. This is confirmed by other studies [27,28].

A significant improvement in the mean preoperative SRS-22r total scores was recorded for both groups. The analyzed SRS-22r questionnaire indicated a clinically and statistically significant improvement from the outcomes before surgical treatment to those of the final follow-up, after surgical correction, in the parameters of self-image and satisfaction for both groups (*p* < 0.001), which are similar to reported studies [7,24,25].

### Limitations

We are aware that our study has some limitations. This study was retrospective in nature and there was not necessarily a sufficient number of patients for comparison; a larger group of subjects would probably be better for showing the differences in the sizes of implants used. Only radiological parameters were analyzed. The assessment of bone union or lack thereof is not always possible on radiological images, and computed tomography would provide better insights and assessment, but exposing patients to an increased dose of radiation is unethical. We consider it advisable to design a prospective, randomized controlled trial to compare the results of different rod sizes to determine which size is better for the surgical treatment of AIS, with a combination of different screw sizes, and the safety of using them. We consider the strength of our study to be the same surgical technique used in all patients and performed by the same two experienced spine surgeons. The meticulous follow-up period and the ability to record all results and complications are also strong points of this study.

## 5. Conclusions

Patients with adolescent idiopathic scoliosis treated with posterior spinal fusion using pedicle screws with an increased diameter and thicker and stiffer rods showed greater correction and better restoration in postoperative thoracic kyphosis without implant failure, with a low risk of loss correction during follow-up and an acceptable risk of complications. The radiographic benefits confirm that increasing the diameter of pedicle screws and connecting them to thicker and stiffer rods for the correction of pediatric spinal deformities is a safe and very effective method and improves health-related quality of life.

## Figures and Tables

**Figure 1 jcm-13-02174-f001:**
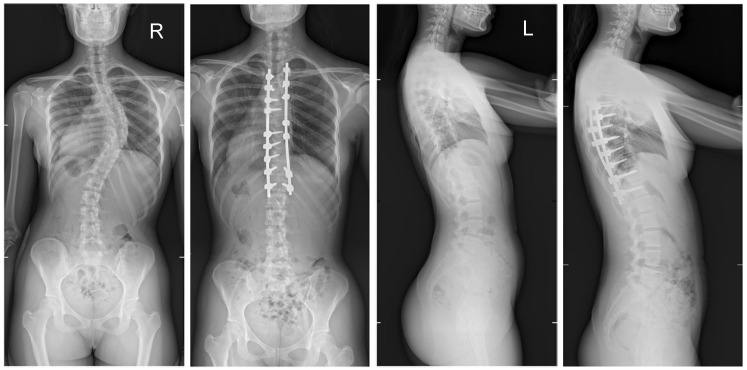
X-rays of an 18-year-old girl with AIS, treated with selective thoracic fusion. A larger screw size of 6.5 mm and 2 cobalt–chromium rods of 6.0 mm were used (G2). X-rays show curve before surgery and correction after surgical treatment.

**Figure 2 jcm-13-02174-f002:**
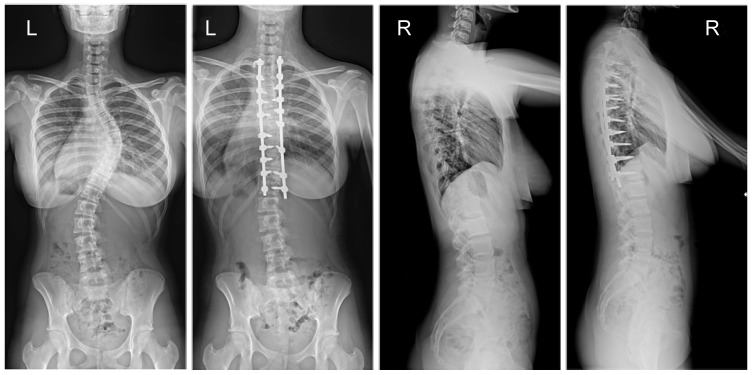
X-rays of a 16-year-old girl with AIS, treated with selective thoracic fusion. A smaller (standard) screw size of 5.5 mm and 2 titanium rods of 5.5 mm (G1) were used. X-rays show curve before surgery and correction after surgical treatment.

**Figure 3 jcm-13-02174-f003:**
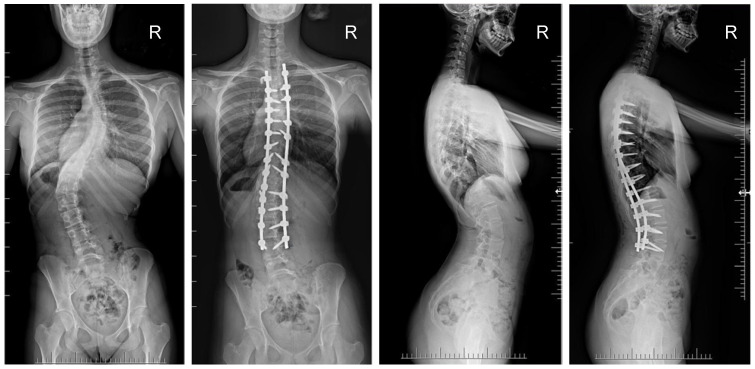
X-rays of a 15-year-old girl with AIS, treated with larger screw sizes of 6.0 and 6.5 mm and 2 cobalt–chromium rods of 6.0 mm (G2). X-rays show curve before surgery and correction after surgical treatment.

**Figure 4 jcm-13-02174-f004:**
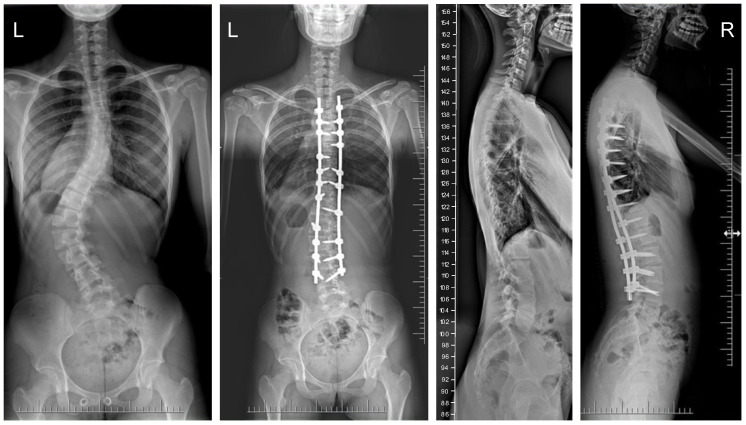
X-rays of a 15-year-old girl with AIS, treated with smaller (standard) screw size of 5.5 and 2 titanium rods of 5.5 mm (G1). X-rays show curve before surgery and correction after surgical treatment.

**Figure 5 jcm-13-02174-f005:**
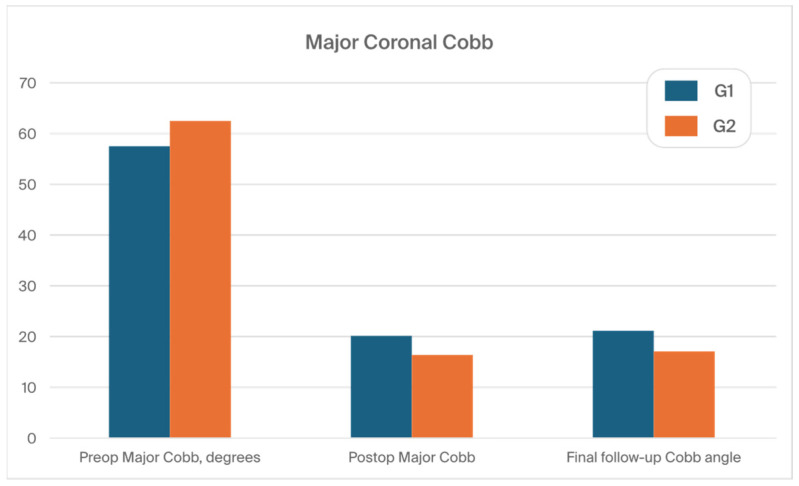
The main curve measures in both groups during the treatment course.

**Figure 6 jcm-13-02174-f006:**
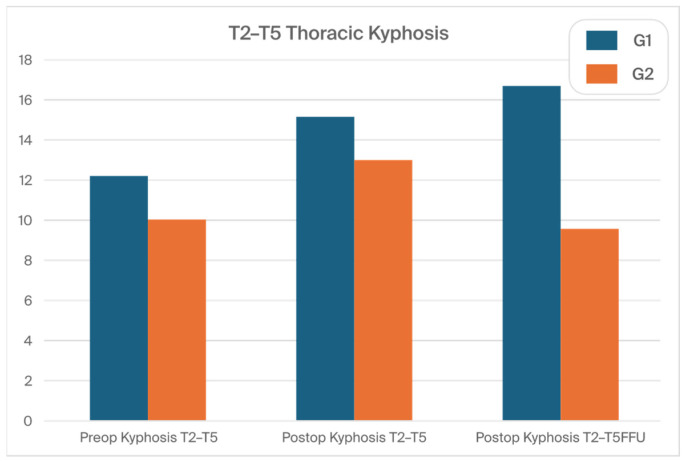
The thoracic kyphosis T2–T5 values in both groups during the treatment course.

**Figure 7 jcm-13-02174-f007:**
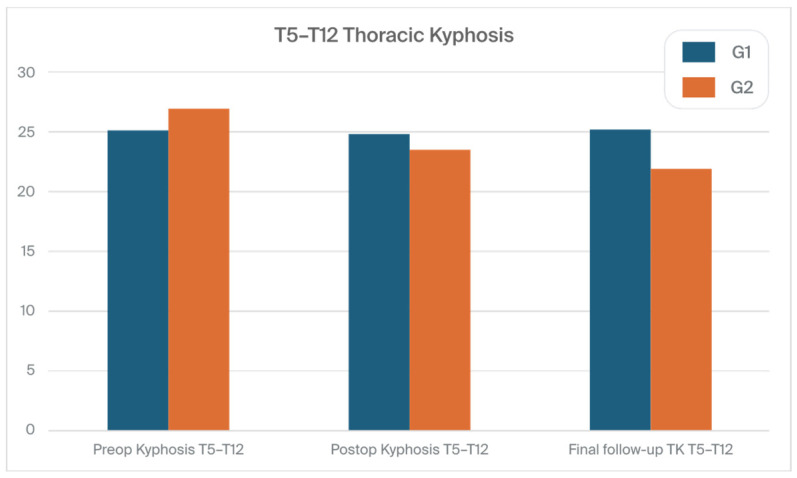
The thoracic kyphosis T5–T12 values in both groups during the treatment course.

**Table 1 jcm-13-02174-t001:** Demographic comparison between G1 and G2 patients. Data all represent mean values and standard deviations for each group; *p*-value was calculated by Fisher’s exact (*) test. *p* < 0.05 was considered statistically significant.

	G1 (*n* = 132)	G2 (*n* = 128)	*p*
Age	14.8 (6.1)	14.8 (5.2)	0.856
Height	161 (8)	162 (9)	0.921
Mean follow-up	44 (8)	42 (9)	0.032
Weight	57 (14)	59 (13)	0.918
Sex (*n*)			
M	13	15	
F	119	113	
BMI	21 (5)	21 (4)	0.992
Risser grade	3.6 (1.8)	3.2 (1.2)	0.979
Time of surgery	252 (62.4)	344 (82.8)	0.443
Blood loss	480 (288)	632 (278)	0.295
Lenke type *			
I	67	68	N.S.
II	27	25	N.S.
III	5	4	N.S.
IV	6	5	N.S.
V	16	14	N.S.
VI	11	12	N.S.
Fusion levels	10.2 (2.8)	9.8 (3.2)	0.781
Screw density%	75 (12)	75 (10)	0.482
Implant complications	3	4	0.783

**Table 2 jcm-13-02174-t002:** Radiographic comparison between G1 and G2 patients. Data are presented as mean values and standard deviations. *p* < 0.05 is considered statistically significant. Flexibility, % (preoperative Cobb angle–preoperative side bending Cobb angle)/preoperative Cobb angle (%). Percentage of correction, % (preoperative Cobb angle–postoperative Cobb angle)/preoperative Cobb angle (%). Statistical analysis of the radiographic parameters of these two groups was performed using the Mann–Whitney rank sum test, and the unpaired *t* test.

Variable	G1 (*n* = 132)	G2 (*n* = 128)	*p*
Preop major Cobb, degrees	57.6 (11.9)	62.5 (12.1)	0.949
Postop major Cobb	19.8 (10.2)	16.6 (8.2)	0.001
Final follow-up Cobb angle	21.3 (9.2)	17.1 (7.2)	0.001
% Cobb correction, postoperative	69.8 (12.8)	74.5 (11.2)	0.001
% Correction (FFU)	61% (15)	68% (14)	0.001
Flexibility rate	35% (11)	33% (8)	0.382
Ponte osteotomy levels (*n*)	5.8 (2.2)	5.6 (2.5)	0.978
Preop kyphosis T5–T12	25.1 (15.5)	26.9 (15.8)	0.779
Postop kyphosis T5–T12	24.8 (8.8)	23.5 (10.8)	0.012
Final follow-up TK T5–T12	25.2 (9.8)	21.9 (10.2)	0.001
Preop kyphosis T2–T5	12.2 (8.8)	10 (8.9)	0.289
Postop kyphosis T2–T5	15.2 (9.8)	13 (5.9)	0.028
Postop kyphosis T2-T5 (FFU)	16.7 (9.2)	9.6 (6.8)	0.001
Preop kyphosis T2–T12	31.2 (14.8)	33.5 (13.6)	0.676
Postop kyphosis T2–T12	33.1 (11.8)	31.4 (12.6)	0.539
Postop kyphosis T2–T12 (FFU)	34.3 (10.7)	35 (9.8)	0.195
Preop lumbar lordosis	50.8 (10.2)	50.2 (9.6)	0.487
Postop lumbar lordosis	44.2 (12.4)	46.4 (11.8)	0.946
Final follow-up lumbar lordosis	50.2 (11.8)	49.5 (12.3)	0.126
Preoperative apical vertebral translation, mm	67 (15.5)	69 (13.8)	0.282
Postoperative apical vertebral translation, mm	21 (6.2)	20.8 (5.2)	0.081
Apical vertebral translation at final follow-up, mm	24 (5.8)	22 (4.8)	0.001
Preop sagittal balance, mm	37.4 (27.8)	39.9 (28.9)	0.215
Postop sagittal balance, mm	39.8 (28.9)	26.8 (23.1)	0.112
Final follow-up sagittal balance, mm	36.6 (22.6)	27.6 (21.4)	0.001
Preoperative coronal balance, mm	18.5 (11.4)	19.2 (11.8)	0.212
Postoperative coronal balance, mm	8.8 (5.6)	8.1 (6.9)	0.731
FFU coronal balance, mm	9.8 (6.2)	10.6 (7.4)	0.493
Loss of correction, degree	5.9 (3.6)	3.2 (3.2)	0.001

**Table 3 jcm-13-02174-t003:** SRS 22r outcomes in both groups.

SRS-22R	G1 (*n* = 132)			G2 (*n* = 128)			
Parameter	Preoperative	Final Follow-Up	*p*-Values *	Preoperative	Final Follow-Up	*p*-Values	*p*-Values SG vs. LG at FFU
Function	4.32 (0.62)	4.62 (0.40)	0.203	4.12 (0.68)	4.52 (0.52)	0.309	0.632
Pain	4.02 (0.65)	4.28 (0.62)	0.178	3.98 (0.62)	4.38 (0.68)	0.058	0.872
Self-image	3.76 (0.62)	4.46 (0.42)	<0.001	3.88 (0.58)	4.62 (0.38)	<0.001	0.061
Mental health	4.12 (0.62)	4.32 (0.60)	0.125	4.02 (0.82)	4.22 (0.76)	0.171	0.591
Satisfaction	3.80 (0.76)	4.22 (0.70)	<0.001	3.80 (0.66)	4.26 (0.74)	<0.001	0.942
Total score	3.86 (0.92)	4.36 (0.62)	<0.001	3.82 (0.82)	4.26 (0.78)	<0.001	0.839

Values are mean (SD). * Statistical comparisons were performed using the Kruskal–Wallis test. Data are presented as averages and standard deviations. *p* < 0.05 is considered statistically significant.

**Table 4 jcm-13-02174-t004:** Rate of complications following posterior final fusion.

Complication Rates Following Posterior Final Fusion	G1 (*n* = 132)	G2 (*n* = 128)	*p*
Intraoperative neuromonitoring changes	5 (3.8%)	7 (5.5%)	<0.001
Superficial wound infection	2 (1.5%)	2 (1.5%)	NS
Pneumonia	2 (1.5%)	2 (1.5%)	NS
Paresthesia from the lateral cutaneous nerve of the lower limb	5 (3.8%)	6 (4.7%)	NS
Radiculopathy	2 (1.5%)	3 (1.5%)	NS
Deep infection	2 (1.5%)	2 (1.5%)	NS
Screw misplacement (replacement)	3 (2.3%)	4 (2.3%)	NS
Pneumothorax	2 (1.5%)	2 (1.5%)	NS
Cerebrospinal fluid leakage	3 (2.3%)	2 (1.5%)	NS
Total	26 (19.7%)	30 (23.5%)	<0.001

## Data Availability

The data are contained within the article.

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
