# Peer review of "The Influence of Increased Pedicle Screw Diameter and Thicker Rods on Surgical Results in Adolescents Undergoing Posterior Spinal Fusion for Idiopathic Scoliosis"

_jcm, 2024, doi:10.3390/jcm13082174_

Round 1
Reviewer 1 Report
Comments and Suggestions for Authors
- The introduction effectively sets the stage for the study by highlighting the impact of different implant designs, screws, and rod diameters in spinal deformity surgeries. However, it may benefit from a more detailed discussion on
-
1. Clarify why rods of different materials were used instead of opting for a uniform material. This detail can help readers understand the rationale behind material choice and its impact on the study's findings.
2. Address whether the consideration of complications is less detailed than statistical analyses. It would be beneficial to more thoroughly list reasons why changes in intraoperative neuromonitoring are important to avoid misleading readers into routine use of thicker screws without proper justification.
3. Expand on the biomechanical rationale for using larger pedicle screw diameters and thicker rods, incorporating recent biomechanical studies to strengthen the study's objectives.
4. Delve deeper into how rod stiffness and the material properties of rods (e.g., titanium vs. cobalt–chrome) influence post-surgical recovery, including discussing potential pros and cons.
5. Provide a more detailed analysis of complication rates and intraoperative neuromonitoring changes. Discuss the safety concerns of using larger screw diameters, especially in pediatric populations with smaller pedicles, and strategies for managing potential complications.
Author Response
- Dear Sir,
- Thank you very much for the review and your valuable suggestions.
- We have performed your recommendations.
- The introduction effectively sets the stage for the study by highlighting the impact of different implant designs, screws, and rod diameters in spinal deformity surgeries. However, it may benefit from a more detailed discussion on
- Clarify why rods of different materials were used instead of opting for a uniform material. This detail can help readers understand the rationale behind material choice and its impact on the study's findings.
- Expand on the biomechanical rationale for using larger pedicle screw diameters and thicker rods, incorporating recent biomechanical studies to strengthen the study's objectives.
- We added several sentences in discussion with explanations why the larger screws and rods are better for correction of the spinal deformity.
- Provide a more detailed analysis of complication rates and intraoperative neuromonitoring changes. Discuss the safety concerns of using larger screw diameters, especially in pediatric populations with smaller pedicles, and strategies for managing potential complications.
- We added several sentences in discussion with explanations of complications in both groups our patients.
Delve deeper into how rod stiffness and the material properties of rods (e.g., titanium vs. cobalt–chrome) influence post-surgical recovery, including discussing potential pros and cons.
There are no differences in post-surgical recovery, so we did not put in the text.
Address whether the consideration of complications is less detailed than statistical analyses. It would be beneficial to more thoroughly list reasons why changes in intraoperative neuromonitoring are important to avoid misleading readers into routine use of thicker screws without proper justification.
These are described in discussion, that complication do not depend on rods and screws in our patients. The Neuromonitoring changes also has been explained.
Thank you very much for your review and great comments.
Best wishes,
Paweł Grabala, MD, PhD
Medical University of Bialystok, Poland
Reviewer 2 Report
Comments and Suggestions for Authors
The research aims to compare different implants for AIS surgical treatment. The tile has a repletion of the word “Adolescents” which should be revised. The abstract is structured although the background is too long, and it includes the study design which should only be stated in the methods section; I suggest a shortening with a clearer statement of the objective/aim at the end.
The introduction offers enough background intro the topic.
In the methodology section, the stages of the research are presented in an organised manner. The full name of the spine centres from which the data was collected should be stated. The subsection 2.1 should be divided into 2 subsections, with the addition of “surgical technique”. Nothing is mentioned about the exact exclusion/inclusion criteria of the patients. Please use median/mean rather than averages when explaining the statistical steps. Medians are presented with IQR not with SD; Means are presented with SD. Continuous variables that pass the normality check test (as suggested by the authors by using the T test) are presented with mean±SD. The Shapiro–Wilk test is a normality test used for continuous variables, how it was used for categorical variables? – the overall statistical methods should be better explained.
In the results section Table 1 should be revised in accordance with the above-mentioned statistical methodology. Figures 1-4 are not referred to n written text.
The discussions make some references to scientific literature. This section can also be extended adding some details of technical perspectives of screw fixation in relation to other scientific papers ( 10.1016/j.promfg.2020.03.070 ). Limitations of the study are presented at the end of the section.
The conclusions are adequate.
The references are properly edited but can be extended as suggested above.
Author Response
Dear Sir,
Thank you very much for your review and very valuable suggestions.
We fixed all mistakes, typos and rebuild the manuscript as you suggest.
The research aims to compare different implants for AIS surgical treatment. The tile has a repletion of the word “Adolescents” which should be revised. The abstract is structured although the background is too long, and it includes the study design which should only be stated in the methods section; I suggest a shortening with a clearer statement of the objective/aim at the end.
We have implemented your comments. The title and abstract have been corrected.
The introduction offers enough background intro the topic.
Thank you for your opinion.
In the methodology section, the stages of the research are presented in an organised manner. The full name of the spine centres from which the data was collected should be stated. The subsection 2.1 should be divided into 2 subsections, with the addition of “surgical technique”. Nothing is mentioned about the exact exclusion/inclusion criteria of the patients. Please use median/mean rather than averages when explaining the statistical steps. Medians are presented with IQR not with SD; Means are presented with SD. Continuous variables that pass the normality check test (as suggested by the authors by using the T test) are presented with mean±SD. The Shapiro–Wilk test is a normality test used for continuous variables, how it was used for categorical variables? – the overall statistical methods should be better explained. In the results section Table 1 should be revised in accordance with the above-mentioned statistical methodology.
Thank you very much for catching such important errors.
The methodology section has been revised as recommended. We added a paragraph with surgical technique. We have added explanations under the tables. Statistical tests were properly described.
Figures 1-4 are not referred to n written text.
We added references in text.
The discussions make some references to scientific literature. This section can also be extended adding some details of technical perspectives of screw fixation in relation to other scientific papers ( 10.1016/j.promfg.2020.03.070 ). Limitations of the study are presented at the end of the section.
The conclusions are adequate.
The references are properly edited but can be extended as suggested above.
We added some new references and connected it to discussion.
Once again, we thank you very much for your review and very helpful suggestions.
We hope that now our article will be more valuable for readers.
Best wishes,
Paweł Grabala, MD,PhD
Medical University of Bialystok, Poland